# INVARIANT ATTENTION: PROVABLE CLUSTERING UNDER TRANSFORMATIONS

## ABSTRACT

Attention mechanisms play a crucial role in state-of-the-art vision architectures, enabling them to rapidly identify relationships between distant image patches. Conventional attention mechanisms do not incorporate other structural properties of images, such as invariance to geometric transformations, instead learning these properties from data. In this paper, we introduce a novel mechanism, Invariant Attention, which, like standard attention, captures image similarity, but with the additional guarantee of being agnostic to geometric transformations. We provide theoretical assurance and empirical verification that Invariant Attention is far more successful than standard kernel attention on multi-class, transformed vision data, and illustrate its potential to correctly cluster transformed data with intra-class variation.

## 1 INTRODUCTION

In recent years, *self-attention* (Vaswani et al., 2017; Bahdanau et al., 2014) has emerged as a key component of state-of-the-art vision architectures, enabling vision transformers to efficiently identify relationships between distant image patches (Dosovitskiy et al., 2020). *Nonlocal* representations of this kind are effective because natural images typically contain repeated and related content, even at large distances (Buades et al., 2005; Wang et al., 2018).

The core operation in self-attention is repeated kernel averaging, in which new features are generated as weighted averages of current features. Because averaging brings similar feature vectors closer together, this operation has a tendency to produce *clusters* in the collection of feature vectors. We show this via an experiment in Figure 2 where Kernel attention clusters data points with a relatively small euclidean distance between them as opposed to data points that are farther away. When the Euclidean distance between data points within a class is bounded, Figure 2 shows that kernel attention with the radial basis function (RBF) kernel collapses clusters into their respective means in just 4 iterations. In fact, kernel attention (in which attention weights are produced via an arbitrary kernel) can be interpreted as an instance of the classical *mean*

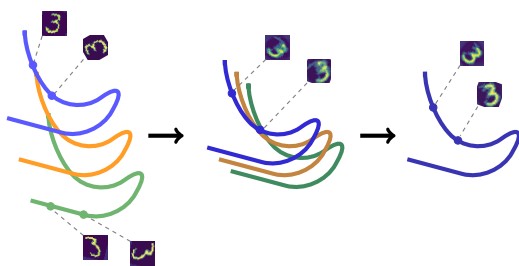

**Figure 1:** *Invariant Attention Achieves Clustering.* Three image manifolds representing three images of the same class with intra-class variation (*left*). After several iterations of Invariant Attention, manifolds get closer to one another (*middle*). Eventually, a cluster center represented by a manifold is found (*right*).

*shift* clustering algorithm by Fukunaga & Hostetler (1975), which has been widely used for image segmentation (Paris & Durand, 2007; Mayer & Greenspan, 2009), tracking (Comaniciu et al., 2000), and other applications.

In transformer architectures, kernel averaging is interleaved with learned transformations, which enable vision transformers to cope with the statistical variability of images, leading to state of the art results on a wide range of tasks (Dosovitskiy et al., 2020; Liu et al., 2021; Lan et al., 2023; Han et al., 2022). These learnable components also allow transformers to neglect certain basic properties of visual data – e.g., the adjacency of the pixel grid (Hinton et al., 2012; Bello et al., 2019) and the

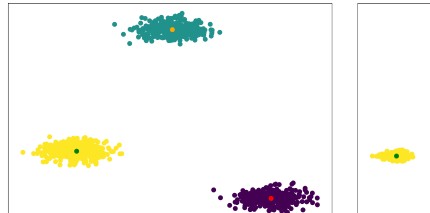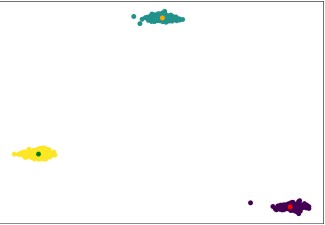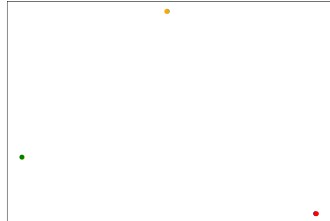

**Figure 2:** *Clustering nature of kernel attention.* Isotropic Gaussian blobs as initial data, and their respective means (left). Blobs after two (middle) and four (right) iterations of kernel attention.

multiscale structure of images (Fan et al., 2021) – at the cost of massive training data and energy. This limitation has inspired a flurry of works which aim to develop hybrid architectures that are effective at capturing nonlocal relationships *and* are insensitive to image translations. Typically, these hybrid methods exhibit significantly improved performance-complexity tradeoffs compared to pure transformers (Bello et al., 2019; Woo et al., 2018; Carion et al., 2020; Wu et al., 2021; Wang et al., 2021; Li et al., 2021; Liu et al., 2021; Fan et al., 2021).

Motivated by the need for data processing strategies which (i) identify relationships across images or image features, and (ii) respect the properties of images, we revisit the core self-attention mechanism from the perspective of transformation invariance. We ask whether it is possible to produce self-attention iterations whose outputs are unchanged, even if the inputs are transformed. We answer this question in the affirmative, analyzing an invariant counterpart to the standard self-attention, which takes as its input a collection of images, image patches, or feature maps, and produces a collection of abstract features, by an invariant form of kernel averaging.

Our main theoretical result shows that this operation rapidly identifies clusters in the data, even if the individual data elements have been transformed. In contrast, standard (transform-variant) clustering methods and standard self-attention lack this property. As a theoretical by-product, we obtain new results on averaging under transformations. Experimental results corroborate our main theoretical claims.

The remainder of this paper is organized as follows: in Section 2, we lay out the basic assumptions and describe the Invariant Attention iteration. Sections 3-4 introduce our main theoretical results, which demonstrate that this iteration clusters transformed data. Section 5 presents simulations and experiments verifying this property. Finally, Section 6 discusses the relationship between our results and the existing literatures on attention, invariance and clustering, and concludes with a discussion of directions for future work.

## 2    FORMULATION: INVARIANT ATTENTION AND KERNEL ATTENTION

**Kernel (Self) Attention.**    Kernel attention (KA) can be represented by the following two steps: (i) we use a kernel $\kappa(\cdot, \cdot)$ to construct a similarity matrix $\mathbf{\Gamma} \in \mathbb{R}^{n \times n}$ and a weight matrix $\boldsymbol{W}$ by column-normalizing $\mathbf{\Gamma}$, and (ii) we replace the current data point $\boldsymbol{v}_j$ with $\boldsymbol{v}_j^+$, the similarity-weighted average of points $\{\boldsymbol{v}_i\}_{i=1}^n$:

$$\boldsymbol{v}_j^+ = \frac{\sum_{i=1}^n \boldsymbol{v}_i \mathbf{\Gamma}_{ij}}{\sum_{l=1}^n \mathbf{\Gamma}_{lj}} = \sum_{i=1}^n \boldsymbol{v}_i \boldsymbol{W}_{ij} \tag{2.1}$$

where $\mathbf{\Gamma}_{ij}$ is the similarity score between images $\boldsymbol{v}_i$ and $\boldsymbol{v}_j$, $1 \le i, j \le n$, and $\boldsymbol{W}_{ij} = \mathbf{\Gamma}_{ij} / \sum_{l=1}^n \mathbf{\Gamma}_{lj}$. Note that $\boldsymbol{v}_j^+$ is the solution to the optimization problem of minimizing sum of squared distances of $\boldsymbol{v}_j$ from all other data points.

The proposed Invariant Attention (IA) also comprises of two steps: (i) finding invariant similarity weights and (ii) finding the quantity called "Invariant Mean" described in detail in the following subsections.

## 2.1 DEFINITIONS RELATED TO TRANSFORMATIONS

Each discrete image (or an image patch[1]) $v_i$ is of size $n_1 \times n_2$ with $n_3$ channels (e.g., $n_3 = 3$ for RGB images), as $v_i \in \mathbb{R}^{n_1 \times n_2 \times n_3}$. We represent a transformation vector field as $\tau \in \mathbb{R}^{n_1 \times n_2 \times 2}$. This field acts on an image $x$, producing a transformed image denoted by $x \circ \tau$, where the pixel $[x \circ \tau]_{pq} = \sum_{kl} x_{kl} \phi(\tau_{pq0} - k)\phi(\tau_{pq1} - l)$. Here, $\phi : \mathbb{R} \to \mathbb{R}$ is the cubic convolution interpolation kernel (Keys, 1981; Buchanan et al., 2022), and $\circ$ denotes function composition. To populate the pixel $(p, q)$ in image $x \circ \tau$, we extract the value located at $(\tau_{pq0}, \tau_{pq1})$ from image $x$.

To compute Invariant Weights and the Invariant Mean (introduced in the next section), the transformation group $\mathbb{T}$ can be any group as long as it satisfies the definition of a group. For example, $\mathbb{T}$ can be the group of affine transformations, perspective transformations etc.

In this paper, for our theoretical and experimental results, we focus on one such transformation group: the Special Euclidean group $\mathbb{T} = \mathrm{SE}(2)$, which can be represented by two parameters: a rotation matrix $A = \begin{bmatrix} \cos\theta & \sin\theta \\ -\sin\theta & \cos\theta \end{bmatrix} \in \mathbb{R}^{2\times2}$ with angle $\theta$ and a translation vector $b \in \mathbb{R}^{2\times1}$. We can write the following relations:

$$\begin{bmatrix} \tau_{pq0} \\ \tau_{pq1} \end{bmatrix} = A \begin{bmatrix} p \\ q \end{bmatrix} + b.$$

This framework can be readily applied to other groups of transformations $\mathbb{T}$. For example, for affine transformations, $A$ must be a non-singular matrix. For similarity transformations (rotation, translation and isotropic scaling), we have an additional dilation parameter. We can then apply gradient descent to optimize these parameters to find the invariant mean.

## 2.2 COMPUTING INVARIANT SIMILARITY WEIGHTS

To compute invariant similarity weights, we construct a kernel that exhibits invariance to geometric transformations. This kernel is denoted by $\kappa^{\max} : \mathbb{R}^d \times \mathbb{R}^d \to \mathbb{R}$, is defined as follows:

$$\kappa^{\max}(v_i, v_j) = \max_{\tau_i \in \mathbb{T}, \tau_j \in \mathbb{T}} \kappa\left(v_i \circ \tau_i, v_j \circ \tau_j\right) \qquad (2.2)$$

The similarity score computed by this kernel is invariant to transformations applied on input images, i.e. $\kappa^{\max}(v, v') = \kappa^{\max}(v \circ \sigma, v' \circ \sigma')$ for some images $v, v'$ and transformations $\sigma, \sigma' \in \mathbb{T}$. The proof for invariance can be found in Section A of the Appendix.

The kernel $\kappa^{\max}$ is not positive definite. However, it *does* provide an invariant similarity measure which (i) can be efficiently computed, and (ii) provably facilitates clustering. Unlike averaging kernels (Mroueh et al., 2015; Haasdonk & Burkhardt, 2007) which achieve invariance through integration, kernel $\kappa^{\max}$ achieves invariance through optimization. This is beneficial for two reasons: first, when the set $\mathbb{T}$ of transformations has moderate dimension (say, 4 dimensions for similarity or 6 dimensions for affine transforms), finding an optimal transformation pair $\tau_i^\star, \tau_j^\star$ is more efficient than averaging over all possible transformations $\tau_i, \tau_j$ in the group $\mathbb{T}$. Second, in many vision applications, the set $\mathbb{T}$ is not a full group, but only a subset of a group.[2] In this setting, averaging kernels are no longer invariant. Nevertheless, our overall Invariant Attention (IA) iteration is compatible with other invariant similarity measures.

While this kernel allows us to compute meaningful similarity scores between images that have been subject to transformations, these scores may not be perfect due to intra-class variation. Consequently, relying solely on this step will be insufficient for accurately identifying clusters within the dataset. As an example of intra-class variation that is affecting similarity scores, consider the MNIST digit "7". It can be depicted with and without a bar across its stem, and the angle between the horizontal and slant lines can vary, among other examples.

---

[1] In the context of this work, the word "image" is used to denote both a complete image and an image patch.

[2] Consider, e.g., similarity transforms, which rotate, translate and scale the image. When working with sampled images, we can only consider scales within a certain range $[s_{\min}, s_{\max}]$. The resulting set $\mathbb{T}$ is not a group, because it is not closed under the group operation.

As in (2.1) for kernel attention (KA), we formulate the weight matrix for Invariant Attention (IA) using an identical procedure. Let the similarity score between images $\boldsymbol{v}_i, \boldsymbol{v}_j$ be given by $\boldsymbol{\gamma}_{ij} = \kappa^{\max}(\boldsymbol{v}_i, \boldsymbol{v}_j)$. We denote the $ij^{\text{th}}$ element of the invariant weight matrix as $\boldsymbol{W}_{ij}$, given by $\boldsymbol{W}_{ij} = \frac{\boldsymbol{\gamma}_{ij}}{\sum_k \boldsymbol{\gamma}_{kj}}$.

Note that the weight $\boldsymbol{W}_{ij}$ tells us how similar images $\boldsymbol{v}_i$ and $\boldsymbol{v}_j$ are *up to transformation*. To find the full similarity matrix $\boldsymbol{W}$, we solve (2.2) for all pairs of images, thus concluding the first step of Invariant Attention (IA).

### 2.3  INVARIANT MEAN FORMULATION

In the first step of IA, we compute correct attention scores for images using the invariant kernel $\kappa^{\max}$. However, relying solely on this step and applying a straightforward weighted averaging of these images, as outlined in KA (2.1), results in a superposition of images that is not meaningful. An example of this is shown in Figure 3b. We therefore propose the Invariant Mean, the solution to the optimization problem [can i put this here? – (2.3)] that aims to find a tensor that is closest on a weighted-average to all the images in such a way that these inherent transformations in each image are inconsequential.

The invariant mean $\boldsymbol{\mu}_j^\star$ [3] is found by optimizing a weighted distance function, given as:

$$\boldsymbol{\mu}_j^\star \quad = \quad \arg \min_{\substack{\boldsymbol{\mu} \\ \boldsymbol{\tau}_1, \ldots, \boldsymbol{\tau}_n}} \sum_{i=1}^n \boldsymbol{W}_{ij} \left\| \boldsymbol{v}_i \circ \boldsymbol{\tau}_i - \boldsymbol{\mu} \right\|_2^2 \tag{2.3}$$

where $\boldsymbol{W}$ is the invariant weights matrix. Note that we jointly minimize over $\boldsymbol{\mu}$ and $\boldsymbol{\tau}_1, \ldots, \boldsymbol{\tau}_n$.

Since $\mathbb{T}$ is a group, by its property of closure under group operation, (i.e. if $\boldsymbol{\tau}, \boldsymbol{\sigma} \in \mathbb{T}$, then $\boldsymbol{\tau} \circ \boldsymbol{\sigma} \in \mathbb{T}$) the solution set to (2.3) is invariant to transformations of input images. Namely, for some transformations $\boldsymbol{\sigma}_1, \ldots, \boldsymbol{\sigma}_n \in \mathbb{T}$ and input data $\bar{\boldsymbol{v}}_1 = \boldsymbol{v}_1 \circ \boldsymbol{\sigma}_1, \ \bar{\boldsymbol{v}}_2 = \boldsymbol{v}_2 \circ \boldsymbol{\sigma}_2, \ \ldots, \ \bar{\boldsymbol{v}}_n = \boldsymbol{v}_n \circ \boldsymbol{\sigma}_n$, we solve precisely the same optimization problem (2.3) as we would for $\boldsymbol{v}_1, \ldots, \boldsymbol{v}_n$. We therefore call $\boldsymbol{\mu}_j^\star$ as the *Invariant* Mean – invariant to transformations present in input data. The corresponding optimal transformations of the two problems, however, are different and are related by the following equations: $\bar{\boldsymbol{\tau}}_1^\star = \boldsymbol{\sigma}_1^{-1} \circ \boldsymbol{\tau}_1^\star, \ \bar{\boldsymbol{\tau}}_2^\star = \boldsymbol{\sigma}_2^{-1} \circ \boldsymbol{\tau}_2^\star, \ \ldots, \ \bar{\boldsymbol{\tau}}_n^\star = \boldsymbol{\sigma}_n^{-1} \circ \boldsymbol{\tau}_n^\star \in \mathbb{T}$. Here $\boldsymbol{\tau}_1^\star, \ldots \boldsymbol{\tau}_n^\star$ are optimal for $\boldsymbol{v}_1 \ldots \boldsymbol{v}_n$, and $\bar{\boldsymbol{\tau}}_1^\star, \ldots \bar{\boldsymbol{\tau}}_n^\star$ are optimal for $\bar{\boldsymbol{v}}_1 \ldots \bar{\boldsymbol{v}}_n$.

Our mechanism's iterative nature arises from replacing the initial image matrix $\boldsymbol{V} = [\ \boldsymbol{v}_1 \mid \boldsymbol{v}_2 \mid \ldots \mid \boldsymbol{v}_n \ ]$ with the updated matrix $\boldsymbol{V}^+$ generated by Invariant Attention. For a given point $\boldsymbol{v}_j$, the next iterate $\boldsymbol{v}_j^+$ is given by setting $\boldsymbol{v}_j^+ \leftarrow \boldsymbol{\mu}_j^\star$, meaning that we replace each image with its Invariant Mean. This concludes one full cycle of Invariant Attention.

For convenience, we define the following function:

$$\varphi(\boldsymbol{\mu}) \quad = \quad \min_{\boldsymbol{\tau}_1, \ldots, \boldsymbol{\tau}_n} \sum_{i=1}^n \boldsymbol{W}_{ij} \left\| \boldsymbol{v}_i \circ \boldsymbol{\tau}_i - \boldsymbol{\mu} \right\|_2^2 \tag{2.4}$$

and note that $\arg \min_{\boldsymbol{\mu}} \varphi(\boldsymbol{\mu})$ is equivalent to the definition of the Invariant Mean in (2.3). To get a closed-form expression for $\boldsymbol{\mu}$, we use the first order optimality condition $\nabla_{\boldsymbol{\mu}} \varphi(\boldsymbol{\mu}) = 0$.

The differentiation of (2.3) to determine the optimal $\boldsymbol{\mu}$ is permissible only when unique optimal transformations $\boldsymbol{\tau}_i^\star$ are used, as per Danskin's theorem. For these transformations to be unique, we must impose certain geometrical conditions and utilize properties of image manifolds. This is explained in detail in the Appendix. Under these conditions, we have that

$$\boldsymbol{\mu}_j^\star = \frac{1}{\sum_{\ell'} W_{\ell' j}} \sum_\ell W_{\ell j} \left( \boldsymbol{v}_\ell \circ \boldsymbol{\tau}_\ell^\star \right), \tag{2.5}$$

Since the Invariant Mean is a weighted sum of images that have been acted upon by their optimal transformations in (2.3), the solution to (2.3) is called invariant "mean".

---

[3] In the context of this work, we use star ($\star$) to indicate optimality.

## 2.4 COMPUTING THE INVARIANT MEAN

To compute the Invariant Mean, we use gaussian smoothing in the optimization problem in (2.3). Since the optimization objective in (2.3) is non-convex, smoothing helps us increase the basin of attraction. Our optimization problem then becomes:

$$
\boldsymbol{\mu}_j^\star \;=\; \arg\min_{\substack{\boldsymbol{\mu} \\ \boldsymbol{\tau}_1,\ldots,\boldsymbol{\tau}_n}} \sum_{i=1}^{n} \boldsymbol{W}_{ij} \left\| g_{\sigma^2} * (\boldsymbol{v}_i \circ \boldsymbol{\tau}_i - \boldsymbol{\mu}) \right\|_2^2 \tag{2.6}
$$

where $\sigma$ is the smoothing level and $*$ represents convolution with a gaussian kernel of the same size. We also denote the following:

$$
f(\boldsymbol{\tau}_1, \boldsymbol{\tau}_2, \ldots, \boldsymbol{\tau}_n; \boldsymbol{\mu}) = \sum_{i=1}^{n} \boldsymbol{W}_{ij} \left\| g_{\sigma^2} * (\boldsymbol{v}_i \circ \boldsymbol{\tau}_i - \boldsymbol{\mu}) \right\|_2^2 \tag{2.7}
$$

To compute the Invariant Mean, we use gradient descent to minimize the objective in (2.7). We begin with $n$ identity transformation vector fields $\boldsymbol{\tau}_1^{(0)}, \boldsymbol{\tau}_2^{(0)}, \ldots, \boldsymbol{\tau}_n^{(0)}$ that correspond to no translation or rotation, each for one of the input images $\boldsymbol{v}_1, \boldsymbol{v}_2, \ldots, \boldsymbol{v}_n$. At the $k^{\text{th}}$ iteration of descent, $\forall i \in \{1, \ldots, n\}$, we compute gradients for $\boldsymbol{\tau}_i^{(k)}$ and corresponding gradients for $\left( \boldsymbol{A}_i^{(k)}, \boldsymbol{b}_i^{(k)} \right)$ using the chain rule, and finally compute gradients with respect to angle $\theta^{(k)}$ using the chain rule. With these updated SE(2) parameters, $\boldsymbol{b}_i^{(k+1)}, \theta_i^{(k+1)}$ compute $\boldsymbol{A}_i^{(k+1)}$, and then $\boldsymbol{\tau}_i^{(k+1)}$ for the next iteration. At each iteration $k$, we then update the Invariant Mean to obtain $\boldsymbol{\mu}^{(k+1)}$ via the following equation based on (2.5):

$$
\boldsymbol{\mu}_j^{(k+1)} = \frac{1}{\sum_\ell \boldsymbol{W}_{\ell j}} \sum_i \boldsymbol{W}_{ij} \boldsymbol{v}_i \circ \boldsymbol{\tau}_i^{(k+1)}.
$$

For a given index $i$, computing the gradients and the interoplated images contain several operations – convolutions are represented by $*$, interpolation is represented by $\circ$, and element-wise multiplication is denoted by $\odot$. We have parallelized these operations across each element using CuPy on NVIDIA T4 GPUs. For example, We note that the gradient of (2.7) is given by $\nabla_{\boldsymbol{\tau}_i} f(\boldsymbol{\tau}_1, \boldsymbol{\tau}_2 \ldots \boldsymbol{\tau}_n; \boldsymbol{\mu}) = 2 g_{\sigma^2} * g_{\sigma^2} * (\boldsymbol{v}_i \circ \boldsymbol{\tau}_i - \boldsymbol{\mu}) \odot \frac{d(\boldsymbol{v}_i \circ \boldsymbol{\tau}_i)}{d\boldsymbol{\tau}_i}$.

We further note that interpolations and gradient calculations for a given index $i$ are independent of other indices. We therefore have further parallelized our implementation across the $n$ transformation vector fields in CuPy.

The Invariant Mean is a feature tensor that we learn from the input images. Algorithm 1 presents pseudo code outlining the optimization procedure for the Invariant Mean.

---

**Algorithm 1** Computing the Invariant Mean

---

**Input** data matrix $\boldsymbol{V} = [\boldsymbol{v}_1, \ldots, \boldsymbol{v}_n]$, learning rate $t$, # of iterations $N$, invariant weights $\boldsymbol{W}_{:j}$

1: Set $[\boldsymbol{\tau}_1^{(0)}, \ldots, \boldsymbol{\tau}_n^{(0)}]$ as a set of identity transformation vector fields.
2: Set the initial invariant mean $\boldsymbol{\mu}_j^{(0)}$ as the sample mean of input data $\boldsymbol{V}^{(0)}$.
3: Set $f(\boldsymbol{\tau}_1, \boldsymbol{\tau}_2 \ldots \boldsymbol{\tau}_n; \boldsymbol{\mu})$ to be the objective as in (2.7).
4: **for** k=0,1, ..., N **do**
5: $\qquad \left[ \boldsymbol{\tau}_1^{(k+1)}, \ldots, \boldsymbol{\tau}_n^{(k+1)} \right] \leftarrow \left[ \boldsymbol{\tau}_1^{(k)}, \ldots, \boldsymbol{\tau}_n^{(k)} \right]$
$\qquad\qquad - t \left[ \nabla_{\boldsymbol{\tau}_1} f\left( \boldsymbol{\tau}_1^{(k)}, \ldots, \boldsymbol{\tau}_n^{(k)}; \boldsymbol{\mu}^{(k)} \right), \ldots, \nabla_{\boldsymbol{\tau}_n} f\left( \boldsymbol{\tau}_1^{(k)}, \ldots, \boldsymbol{\tau}_n^{(k)}; \boldsymbol{\mu}^{(k)} \right) \right]$
6: $\qquad \boldsymbol{\mu}_j^{(k+1)} \leftarrow \frac{1}{\sum_\ell \boldsymbol{W}_{\ell j}} \sum_i \boldsymbol{W}_{ij} \left( \boldsymbol{v}_i \circ \boldsymbol{\tau}_i^{(k+1)} \right)$
7: **end for**
**Output** $\boldsymbol{\mu}_j$

---

## 3 UNIQUENESS OF THE INVARIANT MEAN (UP TO TRANSFORMATIONS)

Now that we are able to compute the invariant mean, we would like to know what would happen when we apply one iteration of Invariant Attention and then use the computed invariant means as inputs for the next iteration and repeat this process many times. We take a mathematical approach to answering this question and give the main results in this section and subsequent Section 4. A detailed and interesting proof is found in the Appendix. We corroborate our proofs with experiments in Section 5.

The proofs and theory in the paper uses a continuum model for images while for experiments and computation we use the discrete formulation of images given so far. This is described in detail in the Appendix Section D. Further, we allow $\mathbb{T} = \text{SE}(2)$ for the theoretical sections of our work.

To prove any result related to clustering, we first have to understand the nature of the solution to the optimization problem in (2.3) which form the data for the next iteration. Equation (2.5) arrived at from the first order optimality equation gives us a closed-form expression of the Invariant Mean but doesn't convey information about second order optimality conditions needed to characterize it as local minima. Furthermore, we are interested in global minima in (2.3).

We try to answer these questions in this section for $\mathbb{T} = \text{SE}(2)$. We recall the norm-preserving property of Euclidean transformations: $\|\boldsymbol{v}\|_2 = \|\boldsymbol{v} \circ \boldsymbol{\tau}\|_2$, $\boldsymbol{\tau} \in \mathbb{T}$. We also recall that $\mathbb{T}$ is a group with composition (denoted $\circ$) as its group operation, so it is closed under transformations. Using these two properties, we can readily show that if $\boldsymbol{\mu}_j^\star$ is a minimizer of (2.3), then so is $\boldsymbol{\mu}_j^\star \circ \boldsymbol{\tau}$ for $\boldsymbol{\tau} \in \mathbb{T}$. We therefore see that our solution set to our optimization problem in (2.3) contains a tensor (image) acted upon by the all transformations of the group. The solution set contains the following set of images which forms a manifold in image space:

$$S_{\boldsymbol{\mu}_j^\star} = \{\boldsymbol{\mu}_j^\star \circ \boldsymbol{\sigma} \mid \boldsymbol{\sigma} \in \mathbb{T}\} \tag{3.1}$$

This manifold has the same dimensions as the dimensions of the transformation group. For $\mathbb{T} = SE(2)$, it is two dimensional. It however remains to prove if the solution set only contains elements of $S_{\boldsymbol{\mu}_j^\star}$.

For each data point $\boldsymbol{v}_1, \ldots, \boldsymbol{v}_n$, we define a corresponding transformation manifold: $S_i = \{\boldsymbol{v}_i \circ \boldsymbol{\tau}_i \mid \boldsymbol{\tau}_i \in \mathbb{T}\} \quad \forall i \in \{0, 1, \ldots, n\}$.

We define the distance between two manifolds defined as

$$d(S_i, S_j) = \min_{\boldsymbol{\tau}_i, \boldsymbol{\tau}_j} \|\boldsymbol{v}_i \circ \boldsymbol{\tau}_i - \boldsymbol{v}_j \circ \boldsymbol{\tau}_j\|_2 \tag{3.2}$$

These are the image transformation manifolds where each point on the manifold represents a base image subject to a transformation. The manifolds of images belonging to the same class and exhibiting intra-class variation have distinct manifolds that are closer (distance given by (D.63)) to each other than they are to manifolds of other classes. For example, the two image transformation manifolds of the digit MNIST 7 written with a bar across its stem and without a bar are closer to each other than to any image transformation manifold of the digit 3. That is, for any two images $\boldsymbol{v}_i, \boldsymbol{v}_j$ that belong to the same class, $d(S_i, S_j)$ is bounded.

We shall prove that when we have transformation manifolds $\{S_1 \ldots S_n\}$ of images $\{\boldsymbol{v}_1 \ldots \boldsymbol{v}_n\}$ that belong to the same class and exhibit intra-class variation such that $d(S_i, S_j)$ is bounded for any $i, j$, under certain geometrical conditions $S_{\boldsymbol{\mu}_j^\star}$ is the unique global minimizer of equation (2.3). That is, $\boldsymbol{\mu}_j^\star$ is unique up to transformations.

**Reach and Curvature for Convex Combinations.** We introduce an important geometrical summary paramter for our analysis called "infimal convex combination reach", denoted by $\rho_{min}$ defined in (B.9). Since the weights sum to 1, we can think of the invariant mean $\boldsymbol{\mu}_j^\star$ in equation (2.5) as a convex combination of vectors belonging to each image transformation manifold. The reach of a manifold is inversely related to its curvature. We require that the curvature of $S_{\boldsymbol{\mu}_j^\star}$ is bounded to be able to prove our results and introduce this parameter which bounds the reach of all possible convex

combinations.This is explained in detail in the Appendix. With this background we give the first result of this paper:

**Theorem 3.1 (Uniqueness of the Invariant Mean)** *Consider data points $\{\boldsymbol{v}_i\}_{i=1}^n$ and their corresponding transformation manifolds $\mathcal{S}_i = \{\boldsymbol{v}_i \circ \boldsymbol{\tau} \mid \boldsymbol{\tau} \in \mathbb{T} = \mathrm{SE}(2)\}$, and let $\rho_{\min}(\boldsymbol{v}_1, \ldots, \boldsymbol{v}_n)$ denote the infimal convex combination reach, defined in (B.9). Consider the optimization problem*

$$\min_{\boldsymbol{\mu} \in L^2(\mathbb{R}^2)} \varphi(\boldsymbol{\mu}) \equiv \sum_{i=1}^n \min_{\boldsymbol{\tau}_i \in \mathbb{T}} \boldsymbol{W}_{ij} \|\boldsymbol{v}_i \circ \boldsymbol{\tau}_i - \boldsymbol{\mu}\|_{L^2}^2, \tag{3.3}$$

*with $\boldsymbol{W}_{ij} \geq 0$ and $\sum_i \boldsymbol{W}_{ij} = 1$. There exists a numerical constant $c > 0$ such that if*

$$\max_{i,j} d(\mathcal{S}_i, \mathcal{S}_j) \leq c\,\rho_{\min}(\boldsymbol{v}_1, \ldots, \boldsymbol{v}_n), \tag{3.4}$$

*then the solution to (3.3) is unique up to transformation, in the sense that for any pair of solutions $\boldsymbol{\mu}^\star, \boldsymbol{\mu}^{\star\prime}$, we have $\boldsymbol{\mu}^{\star\prime} = \boldsymbol{\mu}^\star \circ \boldsymbol{\tau}$ for some $\boldsymbol{\tau} \in \mathbb{T}$.*

The detailed proof of Theorem 3.1 is supported by interesting geometric derivations and is found in Appendix Section D.

## 4 INVARIANT ATTENTION CLUSTERS

As described in Section 2, we would like to perform many iterations of Invariant Attention.

In this section, we give the theorem which that after many iterations, the manifolds converge to their invariant means, i.e., $S_i^{(t)} \simeq S_j^{(t)}$ as $t \to \infty$, where $t$ is the iteration number. This is illustrated in 1. That is, given a set of images from the same class, by showing convergence, we prove that Invariant Attention clusters.

We denote the new image transformation manifold $S_j^+ = \{\boldsymbol{v}_j^+ \circ \boldsymbol{\tau} \mid \boldsymbol{\tau} \in \mathbb{T}\}$ and $d(S_i, S_j) = \min_{\boldsymbol{\tau}_i, \boldsymbol{\tau}_j} \|\boldsymbol{v}_i \circ \boldsymbol{\tau}_i - \boldsymbol{v}_j \circ \boldsymbol{\tau}_j\|_2$.

We now give the second important result of the paper through the following theorem:

**Theorem 4.1** *Let $\boldsymbol{v}_1^{(p)}, \ldots, \boldsymbol{v}_n^{(p)}$ denote the features produced by the $p$-th iteration of Invariant Attention, $\mathcal{S}_j^{(p)} = \left\{\boldsymbol{v}_j^{(p)} \circ \boldsymbol{\tau} \mid \boldsymbol{\tau} \in \mathbb{T}\right\}$ the corresponding transformation manifolds, $\beta$, the rbf kernel bandwidth and*
$R^{(p)} = \max_{m,l} d\left(\mathcal{S}_m^{(p)}, \mathcal{S}_l^{(p)}\right)$.

*There exist positive constants $c, c', \varepsilon$ such that if $R^{(p)} < c\,\rho_{min}(\boldsymbol{v}_1, \ldots, \boldsymbol{v}_n)$ and $\beta < c'/\left(R^{(p)}\right)^2$. Then*

$$d\left(\mathcal{S}_j^{(p+1)}, \mathcal{S}_k^{(p+1)}\right) \leq (1-\varepsilon)d\left(\mathcal{S}_j^{(p)}, \mathcal{S}_k^{(p)}\right). \tag{4.1}$$

The proof for the Theorem 4.1 is found in Appendix Section E. We verify this claim experimentally in Section 5.

## 5 EXPERIMENTS

In this section we show that experimental results corroborate our theory. We see that Invariant Attention indeed clusters invariantly! Our optimization framework for transformation fields follows Buchanan et al. (2022).

**Experiment 1: Clustering of Images.** In this experiment, we have base images of three classes: a hand-drawn crab, a handmade 7 and an image of Pikachu. We then generate 6 transformed images per class. Images are subject to Euclidean transformations with the rotation parameter sampled uniformly at random from the range [-65, 65] degrees, and the translation parameter sampled uniformly at random from the range [-10, 10]. The dataset is shown in the Figure 3a.

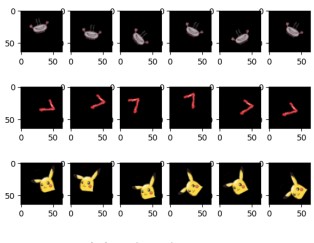 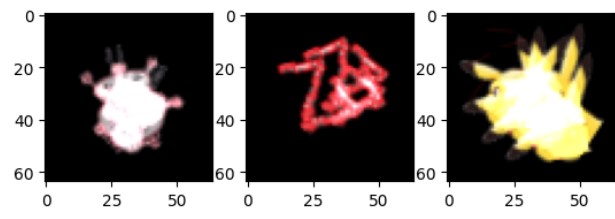

**(a)** The dataset

**(b)** Means of each class calculated by kernel attention

**Figure 3:** *Dataset of three classes.* A dataset of three classes used in experiments (left). Means calculated by kernel attention (KA) do not convey meaningful information (right).

The corresponding sample mean of the these classes is shown in the Figure 3b. We can see that the information conveyed by this mean is not very meaningful.

We then perform 2 iterations of Invariant Attention on this set of images. We note that since there is no intra-class variation, we can find the clusters in just one iteration. However to show the case of clustering even when we find suboptimal solutions to our formulation, i.e. not running our numerical optimization perfectly, we choose a low number for optimization iterations over the weights. Specifically, each optimization iteration is a gradient descent step with a step size 1 and kernel bandwith $\beta = 2$. We therefore see the first iteration of Invariant Attention resulting in an imperfect block matrix (Figure 4a). We also show the invariant mean computed for each image in Figure 4b The matrix is improved upon in the second iteration where we see correct clustering. Invariant means for the second iteration are shown in Figure 4c Such an instance of imperfect weights is also found in the case of intra-class variation.

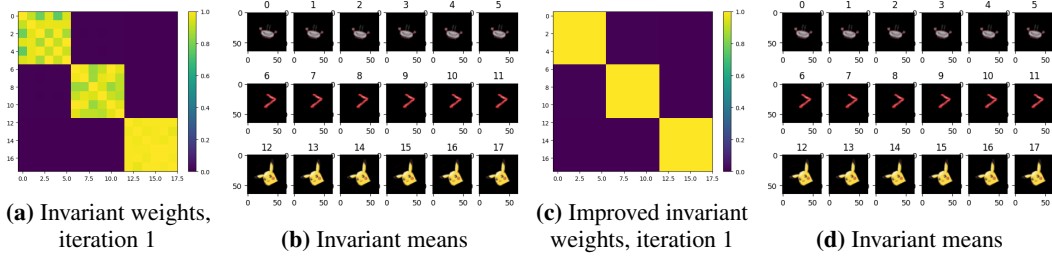

**(a)** Invariant weights, iteration 1

**(b)** Invariant means

**(c)** Improved invariant weights, iteration 1

**(d)** Invariant means

**Figure 4:** *Invariant weights and invariant means* In one iteration, a block-diagonal matrix emerges, showing the desirable Invariant Weights ((a), (c)). Using these Invariant Weights, the Invariant Means of each class are calculated ((b), (d)). Unlike the means calculated by kernel attention in Figure 3b, Invariant Means convey meaningful information.

**Experiment 2: Invariant Mean with Intra-Class Variation.** We show how our Invariant Mean formulation is able to find a meaningful result even in the presence of intra-class variation. We select 10 images of the digit 3 written in different styles from the MNIST dataset (LeCun et al., 1998). These images are subject to rotations chosen uniformly at random within the interval [0, 90] degrees, and a translation also chosen uniformly at random from the interval (0.1, 0.2), which is expressed as the tuple of maximum absolute fraction for horizontal and vertical translations. The dataset for this experiment is shown in Figure 5a. We illustrate the progression of the Invariant Mean by capturing the 0th, 25th, and the 50th iterations of optimization shown in Figures 5b, 5c, and 5d, respectively. We see that at the end of 50 iterations, we have an meaningful Invariant Mean!

**Greedy Search for the Smoothing Parameter.** A constant smoothing value $\sigma$ at every iteration might result in solutions in spurious minima. To avoid this issue and for faster numerical optimization, we do a greedy search over the smoothing level at each iteration of optimization. To do this we enlist a few smoothing levels and choose the one that decreases the invariant mean objective function the most after one step of gradient descent.

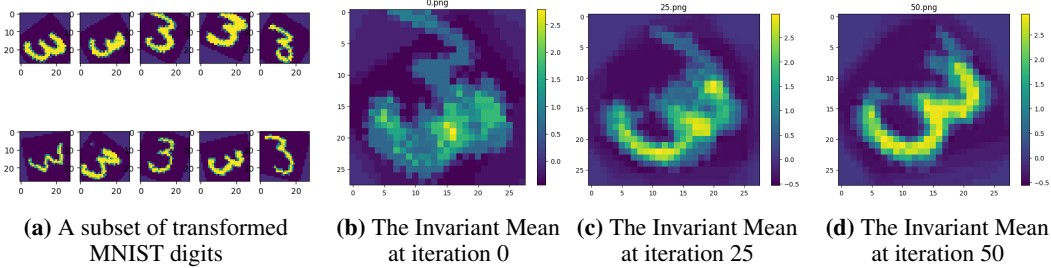

(a) A subset of transformed MNIST digits   (b) The Invariant Mean at iteration 0   (c) The Invariant Mean at iteration 25   (d) The Invariant Mean at iteration 50

**Figure 5:** *Invariant Mean with Intra-Class Variation.* Ten handwritten digits are chosen from the MNIST dataset and are subject to transformations. As the optimization progresses, the Invariant Mean becomes more and more meaningful.

## 6 DISCUSSION AND RELATED WORK

Addressing invariance to geometric transformations in vision is a longstanding challenge. Deep learning has opened empirical avenues to tackle this issue, leading to the development of numerous methods focused on maintaining invariance in images. Several works have focused on learning better feature representations to improve clustering tasks. For instance, Ji et al. (2019) introduced "Invariant Information Clustering," a method that trains a network to predict cluster identities. The central objective of this approach is to learn a representation that preserves commonalities between two data points $x$ and $x'$ by maximizing the mutual information. To capture mutual content, Peng et al. (2019) trained networks to attain invariance by minimizing a KL divergence-based objective, bringing two distributions of latent representations closer to each other. In contrast to seeking improved feature representations, Monnier et al. (2020) proposed to perform clustering in pixel space.

In our work, we introduce Invariant Attention, a novel mechanism designed to embed invariance into an attention model. Central to this approach is the introduction of the invariant kernel $\kappa^{\max}$ – a kernel that determines the highest attainable similarity score between two images by optimizing over transformations. Notably, Liu et al. (2022) also study this kernel, motivated by the challenge of learning with a limited number of samples. Unlike kernels based on averaging, this kernel may not possess positive definiteness. However, this fact does not impede our utilization of it, as our primary objective is to ascertain the maximum similarity between two data points. In this regard, $\kappa^{\max}$ fulfills our intended purpose, regardless of its positive-definiteness. We also introduce the concept of the Invariant Mean, and provide theoretical evidence that Invariant Attention achieves the sought-after property of invariant clustering.

The need for invariance to transformations extends across diverse domains, such as protein structure prediction (Jumper et al., 2021). Within the context of the AlphaFold2 network, the authors introduce a concept termed "Invariant Point Attention" that appears to have a similar name as Invariant Attention proposed in our work. However, Invariant Point Attention in AlphaFold2 aims to maintain invariance under global Euclidean transformations, driven by the fact that the 3D structure of a protein remains consistent regardless of its orientation. In contrast, Invariant Attention enforces invariance under unknown transformations of the domain by optimizing over these transformations.

**Limitations and Future Work,** Following this work, further exploration of the proof of convergence results can be taken up with a particular emphasis on expressing the introduced quantity of infimal convex combination reach in terms of geometric parameters inherent to the problem. Additionally, we envision expanding both the theoretical framework and empirical investigations beyond Euclidean transformations, encompassing similarity and affine transformations and potentially revealing intriguing geometric properties of Invariant Attention. Such developments aim to render Invariant Attention more versatile and applicable in real-world scenarios.

Our overarching goal is to position Invariant Attention (IA) to serve as both a standalone computational primitive and as a modular building block seamlessly incorporated into larger architectural frameworks, including Vision Transformers (ViTs). To achieve this, we are incorporating learnable components within the proposed Invariant Attention module, aimed to enhance Invariant Attention's resilience to statistical variability while simultaneously considering structural properties of images.

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
