# OpenReview forum: "Invariant Attention: Provable Clustering Under Transformations"
_ICLR.cc/2024/Conference — Submitted to ICLR 2024_

### Official Review · Reviewer_4tjm · 2023-10-25

**Soundness:** 2 fair
**Presentation:** 2 fair
**Contribution:** 2 fair
**Rating:** 6
**Confidence:** 3

**Summary:**

The paper presents invariant attention that can cluster images invariant to geometric transformations. It introduces an invariant kernel that computes the maximum similarity between two images after optimizing over transformations. This allows computing meaningful attention weights between transformed images. In addition, the paper presents a theoretical foundation for the approach, demonstrating its efficacy through some simple experiments.

**Strengths:**

1. A new attention mechanism that incorporates invariance properties.
2. The paper provides a solid theoretical foundation for the properties of invariant attention with proof.
3. While the concept of invariance is not a new idea, it remains crucial for the transformer architecture.

**Weaknesses:**

1. While the paper presents mathematical formulations specific to its method, it's not immediately clear how this approach can be adapted or generalized to ViT or other transformer architectures.
2. The proposed method is still based on dynamic kernels [1, 2, 3]. Why the kernels based on averaging are better than previous attempts?
3. Current empirical validation is limited - more quantitative experiments ($e.g.,$ overall accuracy over MNIST) would strengthen the claims. In addition,  more qualitative results on complex image datasets ($e.g.,$ CIFAR100) or the impact of downstream tasks would be useful.
4. (Minor) There are numerous instances of "??", likely due to the separate submission of the main text and supplementary material. Also, there are some typos ($e.g.,$ in theorem 4.1, " invariant attention"). The authors should proofread carefully.


[1] Spatial Transformer Networks.

[2] LocalViT: Bringing Locality to Vision Transformers.

[3] Learning from Few Samples: Transformation-Invariant SVMs with Composition and Locality at Multiple Scales.

**Questions:**

Please see the weaknesses.

---

> ### Author Response · Authors · 2023-11-23
>
> Dear Reviewer 4tjm, thank you for your valuable feedback. Below we respond to your comments in detail.
>
> 1. We thank you for this important remark. We have improved the text to make the motivation and approach more clear, while acknowledging that incorporating IA into a ViT is yet to be shown. However, we believe that the following intuitions hold promise. One of the reasons that transformers require large amounts of data compared to CNNs to achieve the same accuracy is that transformers ignore structural properties of images, especially geometric and spatial properties. Images naturally contain repeated motifs that are some geometric transformation of each other due to factors such as symmetry, camera motion/perspective, object orientation, etc. These geometric transformations are combined with statistical variability to produce intra-class variability in these motifs. CNNs are equipped with the geometric property of translation-equivariance. Since attention mechanisms in transformers offer no such benefits, in our work, we propose Invariant Attention (IA) as a potential replacement to the kernel (self) attention sublayer in a ViT layer while keeping the feedforward sublayer intact. We believe that, enhanced with IA, the transformer architecture would be invariant to geometrical transformations in data, thus eliminating the need to learn invariance from scratch and reducing the total amount of data needed to train a Vision Transformer. Attention mechanisms possess intrinsic clustering properties, as elucidated in the Introduction and visually represented in Figure 2. This is particularly evident with image data that contain motifs subject to geometric transformations. In such cases, the Euclidean distance between transformed images belonging to the same class would not be small, thereby incorrectly clustering upon repeated application of kernel attention. It is this clustering nature of attention that we have tried to address through a generalized optimization framework, and provided a proof for one group of transformations. We believe this inherent clustering property of Invariant Attention will introduce benefits of transformation-invariance to transformers.
>
> 2. Thank you for pointing out these excellent references. These works propose valuable ideas for endowing neural networks with invariance [1] and/or locality [2, 3]. For example, the authors of Spatial Transformers [1] present a learnable module that achieves invariance to transformations via a so-called localization network (a fully-connected network or a CNN). The authors in [2] present LocalViT, where locality is added to vision transformers via adding depth-wise convolution into the feed-forward network. Both works use approaches that take advantage of the 2D structure of images (e.g. adding convolutions). While these works provide substantial empirical results, they remain a black-box approach, and do not aim to provide theoretical analysis that would provide further insight or performance guarantees. Additionally, none of them propose an attention-mechanism based method. In contrast, the proposed Invariant Attention mainly provides theoretical results of provable clustering and simple proof-of-concept experiments that do not involve training. We agree that [1-3] are notable works in the field. Lastly, our proposed kernel is not based on averaging but on maximum similarity. The authors in [3] also propose such a kernel and use probabilistic analysis to show its strengths and valuable properties, e.g. positive definiteness with high probability. In our proposed Invariant Attention, however, we utilize this kernel to calculate invariant weights.
>
> 3. Thank you for the constructive feedback on the limitations of empirical analysis. We acknowledge this need and are working to provide more quantitative experiments over MNIST. However, to run experiments on CIFAR100 or other datasets of natural images where statistical variability takes many forms other than geometry, one would need to incorporate a more complex model (e.g. a network with nonlinearities), which we leave for future directions of this work.

---

### Official Review · Reviewer_9EKX · 2023-10-30

**Soundness:** 2 fair
**Presentation:** 2 fair
**Contribution:** 2 fair
**Rating:** 3
**Confidence:** 3

**Summary:**

The paper proposes a method for calculating attention between images (or image patches), which is invariant to some pre-defined set of transformations. The attention mechanism itself is formulated as kernel attention, and the kernel is constructed to be invariant to a set of transformations. The paper states that iteratively applying their attention mechanism clusters the provided samples into their invariant means, and provides some theoretical guarantees of convergence of this procedure. Authors claim that their findings could help build novel, data-efficient, invariant attention mechanisms implemented into modern vision Transformer networks.

**Strengths:**

The proposed method has a nice clustering quality, in which it tends to cluster similar images together. This feature is proved theoretically and empirically. However, the proof is questionable.

**Weaknesses:**

The paper is very poorly written and looks raw. Apart from the large number of typos, and repetitions (which I will list separately), there are some major issues with the statements themselves.

1. The problem of the paper is not clear to me. What are the problems where the Invariant Attention is needed? Will it increase the overall data efficiency of ViT-type models? How is the clustering property helpful in this case?
2. There are no proofs of the theorems and claims, and not even a sketch of the proof or an idea is provided. Though the authors claim to put it in the appendix, it is not possible right now to review the correctness. The definition of “infimal convex combination reach”, which is an important part of the theory, is not provided even on an intuitive level.
3. At the same time, there is a lot of redundancy. Some almost self-evident claims are explained in long, like the fact that the invariant mean is indeed invariant under image transformations, which is evident from its close form.
4. It is not clearly stated which transformations are admissible. Authors claim that their method works for any transformation set, but provide theoretical guarantees only in the case of the T=SE(2). At the same time, they state that Invariant Attention enforces invariance under unknown transformations of the domain, which is clearly misleading, and the transformation set should be known beforehand to construct the kernel.
5. The optimization procedure for finding invariant mean is not fully described. How are the transformation vectors \tau_i parametrized in each experiment? Exactly, what parameters are we optimizing, and how? It would be nice to have a clearly described algorithm in the form of pseudo-code or something like that. Also, no information about the time complexity or needed resources is provided in the experiment part.
6. Not a learnable algorithm. Though the authors claim that they are currently working at implementing learnable weights inside Invariant Attention, its real applicability to modern visual transformer models in the presented form is questionable. It requires running an optimization procedure for each attention head and each pair of image patches only to calculate the kernel weights. This is also dependent on the dimension and complexity of the transformation set and will require training separate models for different symmetry groups. The issue is not addressed in the paper.
7. Novelty. The kernel attention mechanism was earlier introduced by (Tsai et. al., 2019), and the kernel used in the calculations was described by (Liu et.al, 2021), so the only novel part is in the theoretical results of the paper, which are not significant enough. It is no wonder that we will identify clusters in the data when the data itself is composed of the groups of samples varied through transformations, and we seek to find these exact transformations to match two samples.

The typos:

1. Page 5: “(distance given by (??))”, “This is described in detail in the appendix section ??”, “As illustrated in Figure ??, we have that”. Also, $\phi(\mu)$ is not defined beforehand here.
2. Page 6: “The definitions and its motivations are found in appendix section ??”, “and is found in Appendix Section ??”
3. The $\beta$ in theorem 4.1 is not defined.
4. Page 7: “We see that at the end of 50 iterations, we have an meaningful invariant mean!”
5. Page 8: The subtitle “Figure 4: Invariant weights and invariant means of” is not complete. Also, the same for Figure 5.
6. Page 9: there are 2 almost exact paragraphs on the Invariant Point Attention

**Questions:**

1. Could you describe possible applications of the Invariant Attention for real-world data?
2. Your results (Theorem 3.1) are formulated for SE(2) explicitly. How is that transferable to other groups of transformations?
3. What kind of structural properties are preserved or exploited by Invariant Attention? How will it help in prediction? May invariance to transformations actually harm the prediction quality, when the focus is on orientation, for example? Like classifying the right arrow and left arrow, for example.

---

> ### Author Response · Authors · 2023-11-23
>
> Responses to Weaknesses:
> 1. Thanks for the comment! Clustering is an important problem in unsupervised machine learning. Currently there is no provable clustering technique for data that contains motifs subject to transformations. We have formulated a framework useful for this task for any general transformation group and have proved clustering for a certain class of transformations, the SE(2) group. Application to ViTs: A reason for transformers needing more data than CNNs to train up to the same level of accuracy is that transformers ignore structural properties of image data, especially geometric and spatial properties. Image patches have repeated motifs that are some geometric transformations of each other due to factors such as symmetry, camera motion and perspective, object orientation, etc. These geometric transformations are combined with statistical variability to produce intra-class variability in these motifs. CNNs are equipped with the geometrical property of translation equivariance. Since attention mechanisms in transformers offer no such benefits, in our work, we propose Invariant Attention (IA) as a potential replacement to the kernel attention (self-attention) sublayer in a ViT layer while keeping the feedforward sublayer intact. We believe that, enhanced with IA, the transformer architecture would be invariant to geometrical transformations in data and additional data that was otherwise needed by the transformer to learn this invariance will no longer be required. This, we believe, would reduce the total amount of data needed to train a Vision Transformer. Attention mechanisms have inherent clustering properties as we have explained in the introduction Section and through experiment in Figure 2. when our data contains images/image patches containing motifs subject to geometric transformations since Euclidean distance between transformed images of the same class is not small would not be clustered together on repeated application of Kernel attention. It is this clustering nature of attention that we have tried to address through a generalized optimization framework and proved theoretically for one group of transformations. We believe this inherent clustering property of Invariant Attention will introduce benefits of transformation-invariance to transformers.
>
> 2. Thanks for the suggestion. The proof involves many geometric ideas, and we could not sketch them in the main body due to space limitations. However, we have improved the Section on Uniqueness, Section 3 to better motivate some ideas leading up to the main results.
>
> 3. Thanks for the comment. From the formulation in equation 2.3, we have used 1. closure properties of a transformation group and norm preserving property of Euclidean transforms to remark that the optimization problem in Equation 2.3 contains solutions that are transformations of each other. This is also not readily evident for other groups of transformations. The closed form expression of the invariant mean also requires additional geometric properties of the data as required by Danskin's theorem and we have highlighted them in the updated version to signify the importance of these observations.
>
> 4. Thank you for the feedback. We have added a new Section, Section 2.1 that details the optimization of transformations and discusses the generalization ability of this framework to other transformation groups We can impart invariance to the group of transformations we desire. Some examples of transformation groups that are geometric transformation groups are Affine group, Similarity group, etc. When implemented for these groups, invariant attention enforces the Invariance property to the domain of the selected transformation group. We have added to text to distinguish the generalizability of the framework to other transformation groups and the transformation group we have chosen for our proof.
>
> 5. Thank you for the feedback. We have added two Sections: Section 2.1 and 2.4 that detail the optimization procedure. In these Sections we have included an algorithm box, optimization procedure and parallelizable nature of our computations.
>
> 6. Thanks for the feedback. When compared to the current Kernel Attention mechanism, Invariant Attention does add computational overhead. However, as described in Section 2.4, most of this overhead is parallelizable and the benefits of invariance properties to transformations can be imparted for clustering or into ViT architectures.

---

> ### Author Response · Authors · 2023-11-23
>
> Response to Questions:
>
> 2. Thank you for the question. We have added an new Section, Section 2.1 that details the optimization of transformations and discussues the generalizability of this framework. Some examples of transformation groups that are geometric transformation groups are affine group, similarity group, etc.
>
> 3. Invariant attention will be helpful for problems in classification where the class label is invariant to transformations. For problems in which orientation is of interest, one solution is to extend the output of IA to output not only the invariant mean, but also the optimal transformations which generated the Invariant Mean. We note that these transformations and the Invariant Mean are jointly optimized in the computation.

---

### Official Review · Reviewer_omyp · 2023-10-31

**Soundness:** 2 fair
**Presentation:** 1 poor
**Contribution:** 2 fair
**Rating:** 3
**Confidence:** 4

**Summary:**

The paper proposes an addition to self-attention that is invariant to various transformations. Mainly, while self-attention is based on the similarity between two entities, the proposed method is based on the maximum similarity between transformed samples. Essentially, the framework proposes replacing $k(x,y)$ with $max_{T_1,T_2} k(T_1(x),T_2(y))$. This way, any transformation $T$ applied to samples $x,y$ does not influence the similarity. Additional non-linearities or learnable parameters are ignored. Two results are proved: 1. The proposed invariant attention results in a unique solution (up to transformation) 2. The procedure converges.

**Strengths:**

Strong Points:

- The invariance of machine learning models is an important topic, useful for generalization and sample efficiency.
- The method seems technically correct.

**Weaknesses:**

Weak Points:

- There are no details on how to obtain the optimal transformations from 2.6. How to obtain these transformations is crucial. Without an efficient way to obtain them, the proposed method cannot be applied in practice.

- The method does not involve actual feature learning. It's hard to argue the importance of the method for machine learning methods when there is no actual representation learning happening.

- The experiments are extremely simple: 6 transformations of the same image or 10 MNIST samples. These might be good for a first step to see that the method/implementation is sound, but more validations are needed for a novel machine learning method.

- “Invariant Attention, enforces invariance under unknown transformations of the domain by optimizing over these transformations” How general are the transformations that Invarian Attention can optimize over? What kind of transformations can be optimized in practice?

**Questions:**

Typo? It seems like equation 3.1 needs two indices, one for sample $v_i$ and one for transformation $\tau_j$. Also, the number of samples and number of transformations should be different.

Minor: There are some broken references. E.g “distance given by (??))”

---

> ### Author Response · Authors · 2023-11-23
>
> Responses to Weaknesses:
> 1. Thank you for the feedback. We have added two Sections: Section 2.1 and 2.4 that detail the optimization procedure.
> 2. Thanks for the comment! Clustering is an important problem in unsupervised machine learning. Currently there is no provable clustering technique for data that contains motifs subject to transformations. We have formulated a framework useful for this task for any general transformation group and have proved clustering for a certain class of transformations, the SE (2) group. We have added details about the optimization procedure for the invariant mean in Section 2.4.
> 3. Thank you for the constructive feedback on the limitations of empirical analysis. We acknowledge this need and are working to provide more quantitative experiments over MNIST. However, to run experiments on CIFAR100 or other datasets of natural iamges where statistical variability takes many forms other than geometry, one would need to incorporate a more complex model (e.g. a network with nonlinearities), which we leave for future directions of this work.
> 4. Thank you for the question. We have added a new Section, Section 2.1 that details the optimization of transformations and discusses the generalizability of this framework. Some examples of transformation groups that are geometric transformation groups are affine group, similarity group, etc.

---

### Official Review · Reviewer_pRB3 · 2023-11-06

**Soundness:** 2 fair
**Presentation:** 2 fair
**Contribution:** 2 fair
**Rating:** 5
**Confidence:** 3

**Summary:**

This paper presents a new attention mechanism called invariant attention. The paper shows that the proposed attention has theoretical guarantees and can be applied to solve image clustering problems.

**Strengths:**

-This paper aims to improve the attention mechanism, which is the foundation in the widely used transformer architecture.

-This paper provides extensive theoretical analysis for the proposed attention mechanism.

**Weaknesses:**

-The experiments are not sufficient to support the claims. First, there is no comparison with previous works in the experimental section. Second, there is no quantitative result. Without those, I cannot judge the if the proposed technic is useful or not and the significance of the proposed method.

**Questions:**

-I cannot find the theoretical proof that shows that "the Invariant Attention is far more successful than standard kernel attention". I might miss this part because I am not an expert in theoretical ML.

---

> ### Author Response · Authors · 2023-11-23
>
> Response to Weaknesses:
>
> Thank you for your constructive feedback. We acknowledge the need for quantitative results, and are currently performing a quantitative comparison of kernel attention and Invariant Attention to measure clustering accuracy. This experiment will be added to the next version of the paper.

---

### Author Response · Authors · 2023-11-23

Dear Reviewers,

We thank you for your valuable and constructive feedback, and the time you have put in reviewing our work. We provide detailed answers to each individual comment below.

---

### Meta-Review · Area_Chair_jXXE · 2023-12-11

**Metareview:**

The paper introduces a novel clustering method for features, akin to spatial attention, but uniquely designed to be invariant to spatial transformations. This innovation aims to enhance the robustness of feature clustering against geometric variances. However, reviewers have reached a consensus that the paper currently falls short of the standards required for publication. Critically, the paper lacks comprehensive empirical evidence and experiments to substantiate its claims. Additionally, the manuscript suffers from a lack of clarity and detail, particularly in explaining the process of deriving optimal transformations. There are also concerns regarding the inadequacy of the presented proofs, which lack the necessary depth and lucidity. Moreover, the paper would benefit significantly from more rigorous comparative analyses, especially with approaches like kernel attention. Given these substantial gaps, it is evident that the paper requires significant revisions and improvements before it can be considered for resubmission.

**Justification For Why Not Higher Score:**

Lack of proper empirical justification.
Lack of clarity.

**Justification For Why Not Lower Score:**

See above

---

### Decision · Program_Chairs · 2024-01-16

Reject